# Knowledge of diabetes and associated factors in rural Eastern Cape, South Africa: A cross sectional study

**Eyitayo Omolara Owolabi**[1]*, **Daniel Ter Goon**[2], **Anthony Idowu Ajayi**[3], **Oladele Vincent Adeniyi**[4]

**1** Faculty of Medicine and Health Sciences, Department of Global Health, Centre for Global Surgery, Stellenbosch University, Tygerberg, Cape Town, South Africa, **2** Faculty of Health Sciences, Department of Public Health, University of Fort Hare, East London, South Africa, **3** Population Dynamics and Sexual and Reproductive Health Unit, Africa Population and Health Research Center, Nairobi, Kenya, **4** Faculty of Health Sciences, Department of Family Medicine, Walter Sisulu University/Cecilia Makiwane Hospital, East London, South Africa

* owolabiomolara101@gmail.com

## Abstract

### Background

Diabetes management is complex and requires several lifestyle modifications and engagement in self-management behaviours to prevent complications and to improve health outcomes. Adequate disease knowledge is required in order to engage in appropriate self-management behaviours. Yet, patients' knowledge of diabetes and associated factors are scarcely investigated. Context-specific data on diabetes knowledge are crucial for designing appropriate interventions for improving knowledge and treatment outcomes. This study examined the level of diabetes knowledge and its associated factors among persons with diabetes in Eastern Cape Province, South Africa.

### Methods

We conducted a cross-sectional study among 399 individuals attending diabetes care at six randomly selected primary healthcare facilities in Eastern Cape. Demographic data were obtained through questionnaire interviews while diabetes knowledge was assessed using the validated Michigan Diabetes Knowledge Test questionnaire. Descriptive and inferential statistics were used to assess the mean diabetes knowledge and its associated factors.

### Results

Participants' median age was 63 (IQR: 54–70) years, and the median diabetes duration was 6 (IQR: 3–13) years. From a total score of 20, participants' knowledge of diabetes ranged from 0 to 17 with an average score of 7.5 (SD±2.2). After controlling for relevant covariates, being employed was positively associated with higher diabetes knowledge (p<0.001). However, health facility level was negatively associated with diabetes knowledge (p = 0.001). Participants receiving care at the community healthcare centres had a lower level of diabetes knowledge compared to those receiving care at the primary healthcare clinics.

**Data Availability Statement:** All relevant data are within the paper. In addition, the study dataset has been previously published as a supplementary file

by Owolabi et al. (DOI: 10.1371/journal.pone.
0224791).

**Funding:** The author(s) received no specific
funding for this work.

**Competing interests:** The authors have declared
that no competing interests exist.

## Conclusion

There was a low level of knowledge on the various components of diabetes management
among individuals with diabetes in this setting. Therefore, context-specific interventions to
improve the knowledge of diabetes is required and should target unemployed individuals
and the community health centres in the region.

## Introduction

Globally, diabetes mellitus (DM) and its related complications constitute significant public
health challenges. There is a growing prevalence of DM globally. In 2019, an estimated 463
million individuals were living with DM globally compared to 211 million in 1990 [1,2], indi-
cating a 120% increase. Most (79%) of these individuals reside in low and middle-income
countries (LMICs) where health resources are scarce and healthcare quality is sub-optimal
[2,3]. In South Africa, there is a huge burden of DM with 4.5 million adults living with DM in
2019 [2,4]. The increasing prevalence of DM in SA is largely driven by an increase in obesity,
unhealthy lifestyle behaviours, an ageing population and urbanisation [4]. Furthermore, DM is
currently among the ten leading causes of death in SA and over half of the deaths occur among
those of the working-age group [4]. The burden of DM in SA is further complicated by a high
level of poorly controlled glycaemic status with rates ranging from 69.3% [5] to as high as 84%
[6], which predispose DM patients to various microvascular and macrovascular complications
and increased healthcare costs.

One of the crucial steps towards reducing DM burden and improving health outcomes is
by promoting lifestyle modifications and patients' engagement in self-management behav-
iours. The integrated theory of behavioural change [7] however posited that positive health
behaviour is fostered through patient education and knowledge. Individuals with DM who are
knowledgeable about DM management can better self-manage their disease condition, easily
identify potential risk factors and will likely seek timely care when needed [8]. This in turn
contributes to better glycaemic status, reduced rate of complications, hospitalisation rate and
improved quality of life [8,9]. It is therefore important to regularly assess patients' knowledge
and to use such data to design appropriate interventions to fill knowledge gaps.

However, data on DM knowledge in SA are limited and differed across provinces and asso-
ciated factors were rarely investigated. Existing studies conducted outside SA have mostly
reported a low to average level of DM knowledge among individuals with DM [8,10–12],
whereas studies in SA have reported conflicting results [13–15]. While Moodley et al [13]
reported good DM knowledge in Kwazulu-Natal, studies in Limpopo and the Free State
reported low levels of knowledge [14,15]. Although, it is worth noting that Moodley et al [13]
considered good knowledge of DM as a score of 50% or higher. A 50% cut off mark do not
necessarily connote a good knowledge. Likewise, associated factors of DM knowledge tended
to differ across various countries. For instance, studies have highlighted a positive correlation
between level of education and diabetes knowledge [16–18]. In addition to level of education,
other studies have also reported a significantly higher DM knowledge among those with a
high-income level, those who had access to diabetes educators or training and among males
[18,19]. Others have shown the significant role of family support and being married on diabe-
tes knowledge and practice [20].

Given that DM knowledge and associated factors vary across settings, there is a need for
context-specific data for designing appropriate interventions. Currently, there is no study on

DM knowledge and its associated factors among persons living with the disease in the Eastern Cape province of SA. This is one of the provinces with the highest prevalence of diabetes, and reportedly low level of glycaemic control [6]. Therefore, this study sought to ascertain the level of knowledge of diabetes and its associated factors among DM patients attending selected primary healthcare centres (PHCs) in the Eastern Cape. Findings from this study will provide baseline data on DM knowledge in this province that may inform the design of appropriate health system responses and interventions.

## Methods

### Study design and setting

This study adopted a descriptive, cross-sectional design to assess diabetes knowledge among patients with DM in Eastern Cape, South Africa. The Eastern Cape Province is one of the poorest provinces in South Africa, with an estimated population of 6.6 million who are predominantly black. The province has two metropolitan municipalities; Nelson Mandela and Buffalo City, and six districts; Alfred Nzo, Sarah Bartman, Amathole, OR Tambo, Joe Nqabi and Chris Hani [21].

### Study population

This study forms part of a larger study that assessed the efficacy, acceptability and feasibility of mHealth in improving adherence to anti-diabetic therapy and glycaemic control [22]. The study population were adults living with DM attending selected primary healthcare centres in Buffalo City Metropolitan Municipality and Amathole Health District, who met the eligibility criteria.

### Sample and sampling technique

The sample size for this study was estimated based on an assumed 50% proportion of those who are knowledgeable about diabetes using the formula:

$$n = z^2 * p * (1-p)/e^2$$

Where: z = 1.96 for a confidence level ($\alpha$) of 95%, p = proportion (expressed as a decimal), e = margin of error.

z = 1.96, p = 0.5, e = 0.05
$n = 1.96^2 * 0.5 * (1-0.5)/0.05^2$
n = 0.9604/0.0025
n = 384

The estimated sample size was 384. The estimated sample size was adjusted in anticipation for incomplete data by a factor of 5%. A total of 404 participants were recruited.

All eligible ambulatory DM patients who were willing to participate were recruited consecutively at the selected PHCs. In this setting, primary healthcare clinics are smaller health facilities staffed by nurses and doctors visiting for limited hours (usually 4–8 hours per week) while community health centres (CHCs) are larger facilities that are staffed by a multidisciplinary PHC team consisting of nurses, doctors, pharmacists and allied health workers. PHCs with specific diabetes clinics were visited on scheduled days, while those who run open clinics were visited daily. Data collection was done for a minimum of two weeks at each clinic.

### Eligibility criteria

The inclusion criteria included participants 18 years and above, confirmed with DM, and attending the selected facilities. Critically ill patients were excluded from the study and directed for acute care management.

### Data collection

Demographic data included age, gender, level of education, employment status and average monthly income, which were obtained using a validated questionnaire via face-to-face interviews. Clinical history, such as duration of illness and presence of comorbidity were obtained by self-reporting and review of clinical records. Participants' knowledge of diabetes was assessed using the simplified Michigan Diabetes Knowledge Test questionnaire, SDKT-2, the true/false version [23]. The SDKT-2 questionnaire contains 20 items in both sections. The first part of the questionnaire had 18 items focusing on patients' general knowledge on diabetes, diets, physical activity and self-efficacy, while the second section had two items on insulin therapy. Every incorrect responses as well as a 'don't know' were considered a knowledge deficit. The Michigan Diabetes knowledge test questionnaire is a validated tool [24] that has been adopted in different countries including South Africa [13,25]. In this study, the diabetes knowledge scale showed high internal consistency. The reliability test of the scale yielded a Cronbach alpha value of 0.94. The questionnaire was administered by a trained, bilingual native speaker who assisted participants in completing the questionnaire in their local language (IsiXhosa).

### Ethical considerations

Ethics approvals for the study were obtained from the University of Fort Hare Research Ethics Committee (Reference number: GOO171OWA01) and the Eastern Cape Departments of Health. In addition, approvals were obtained from the selected health districts managers as well as the clinic managers. Verbal and written informed consents were obtained from the participants prior to the commencement of the study, after due explanation of the research purpose and aim. Rights to anonymity and confidentiality were ensured throughout the study and participants consented to referral to further care in case of detection of abnormal findings.

### Data analysis

Prior to statistical analysis, data were checked for completeness and data from five participants were excluded from the analysis due to incomplete data. Data were expressed as counts (frequency) and proportions (%) for categorical variables, while mean values (±Standard deviation) and median (interquartile range) were used to summarise continuous variables. The mean knowledge difference between variables was determined using ANOVA. Ordinary least squared analysis was used to ascertain relationships between relevant variables and knowledge. A p-value of $< 0.05$ was considered statistically significant. All statistical analyses were performed using STATA 15 (College Park, TX, USA).

### Results

A total of 399 individuals living with diabetes were involved in the study. Table 1 presents the demographic characteristics of the study participants. The median age was 63 years, mean monthly income was 1869 Rands, and median diabetes duration was six years. The majority of the participants were females (81.7%), unemployed (82.2%), had type 2 diabetes (93.2%), and were receiving treatment for hypertension (85.6%).

**Table 1. Demographics characteristics.**

| Variables | n(%) |
|---|---|
| Age, Median (IQR) | 63 (54–70) |
| Income, Mean (SD) | 1869 (1944) |
| Diabetes duration, Median (IQR) | 6 (3–13) |
| Knowledge score, Mean (SD) | 7.5 (2.2) |
| Gender | |
| Male | 73 (18.3) |
| Female | 326 (81.7) |
| Highest level of education | |
| No formal schooling | 10 (2.5) |
| Grade 1–7 | 156 (39.2) |
| Grade 8–12 | 211 (53.0) |
| Tertiary | 21 (5.3) |
| Marital status | |
| Never married | 104 (26.3) |
| Married | 182 (46.1) |
| Divorced | 17 (4.3) |
| Widowed | 91 (23.0) |
| Cohabiting | 1 (0.3) |
| Employment status | |
| Government employee | 8 (2.0) |
| Non-government employee | 15 (3.8) |
| Self-employed | 11 (2.8) |
| Student | 1 (0.3) |
| Retired | 36 (9.0) |
| Unemployed | 328 (82.2) |
| Diabetes type | |
| Type 1 | 27 (6.8) |
| Type 2 | 372 (93.2) |
| Treatment type | |
| Oral pills only | 302 (75.7) |
| Insulin only | 62 (15.5) |
| Both | 35 (8.8) |
| Family history of diabetes | |
| Yes | 196 (49.1) |
| No | 203 (50.8) |
| Hypertensive | |
| Yes | 333 (85.6) |
| No | 56 (14.4) |
| Presence of other comorbidities | |
| Yes | 316 (79.2) |
| No | 83 (20.8) |

On a scale of 0 to 20, participants knowledge of diabetes ranged from 0 to 17 with an average knowledge score of 7.5 (SD±2.2).

As shown in Table 2, participants' knowledge was below average for most of the DM management components. Participants' knowledge of the various dietary aspects of DM differ. Over half were aware of the dangers of fatty food and unhealthy cooking oils, but majority

**Table 2. Descriptive summary of participants' diabetes knowledge.**

| | Correct response n (%) | Incorrect response n (%) |
|---|---|---|
| **Dietary knowledge** | | |
| The diabetes diet is a healthy diet for most people | 232 (58.2) | 167 (41.8) |
| A pound of chicken has more carbohydrate in it than a pound of potatoes | 87 (21.8) | 312 (78.2) |
| Orange juice has more fat in it than low fat milk | 106 (26.6) | 293 (73.4) |
| Unsweetened fruit juice raises blood glucose levels | 133 (33.3) | 266 (66.7) |
| Using olive oil in cooking can help prevent raised cholesterol in the blood | 208 (52.1) | 191 (47.9) |
| Eating foods lower in fat decreases your risk for heart disease | 251 (62.9) | 148 (37.1) |
| **Physical activity** | | |
| Exercising regularly can help reduce high blood pressure | 254 (63.7) | 145 (36.3) |
| For a person in good control exercising has no effect on blood sugar levels | 107 (26.8) | 292 (73.2) |
| **Diabetes testing** | | |
| Glycosylated haemoglobin (HbA1c) is a test that measures your average blood glucose level in the past week | 85 (21.3) | 314 (78.7) |
| Urine testing and blood testing are both equally as good for testing the level of blood glucose | 100 (25.1) | 299 (74.9) |
| **Knowledge of complications** | | |
| Wearing shoes a size bigger than usual helps prevent foot ulcers | 104 (26.1) | 295 (73.9) |
| Having regular check-ups with your doctor can help spot the early signs of diabetes complications | 292 (73.2) | 107 (26.8) |
| Attending your diabetes appointments stops you getting diabetes complications | 66 (16.5) | 333 (83.5) |
| **General knowledge** | | |
| A can of diet soft drink can be used for treating low blood glucose levels | 125 (31.3) | 274 (68.7) |
| Infection is likely to cause an increase in blood sugar levels | 230 (57.6) | 169 (42.4) |
| When you are sick with the flu you should test for glucose more often | 236 (59.2) | 163 (40.8) |
| Numbness and tingling may be symptoms of nerve disease | 198 (49.6) | 201 (50.4) |
| Lung problems are usually associated with having diabetes | 89 (22.3) | 310 (77.7) |
| **Insulin use* (n = 93)** | | |
| High blood glucose levels may be caused by too much insulin. | 19 (20.4) | 74 (79.6) |
| If you take your morning insulin but skip breakfast your blood glucose level will usually decrease. | 58 (62.4) | 35 (36.6) |

lacked knowledge on the caloric content of some common food choices. Only one-fifth of the participants were knowledgeable about diabetes testing using HbA1c. Even though 63.7% of the participants knew that exercise helps to reduce blood pressure, only 26.8% felt it is helpful for blood glucose control.

Pairwise correlation was used to show the relationship between participants' age, income, diagnosis duration and diabetes knowledge. As depicted in Table 3, there was a negative correlation between age and diabetes knowledge while income and diagnosis duration had no relationship with diabetes knowledge. Only ages 18–50, grade 8 and above level of education, never marrying, being employed, family history of DM, not receiving treatment for hypertension, receiving care at primary healthcare clinics were significantly associated with diabetes knowledge.

Table 4 shows the results of the ordinary least square regression analysis. After controlling for relevant covariates, only employment status (p<0.001) and health facility level (p = 0.001) were significantly associated with diabetes knowledge. Employment status was positively and significantly associated with diabetes knowledge which shows that employed participants had

**Table 3. Bivariate analysis showing association between patient characteristics and diabetes knowledge.**

| Variables | Frequency | CC/Mean knowledge (SD) | p-value |
|---|---|---|---|
| Age[a] | 399 | -0.18 | <0.001 |
| Income[a] | 399 | 0.04 | 0.416 |
| Duration of diagnosis[a] | 399 | 0.03 | 0.609 |
| Gender[b] | | | |
| Male | 73 | 7.84 (2.50) | 0.116 |
| Female | 326 | 7.38 (2.13) | |
| Highest level of education[b] | | | |
| No formal education-Grade 7 | 167 | 7.01 (2.16) | <0.001 |
| Grade 8 and above | 232 | 7.80 (2.18) | |
| Marital status[b] | | | |
| Never married | 105 | 7.92 (2.17) | 0.014 |
| Ever married | 290 | 7.30 (2.20) | |
| Employment status[b] | | | |
| Unemployed | 365 | 7.31 (2.04) | <0.001 |
| Employed | 34 | 9.21 (3.02) | |
| Facility level[b] | | | |
| Primary healthcare clinics | 144 | 8.16 (2.21) | <0.001 |
| Community health centre | 255 | 7.08 (2.11) | |
| Family history of diabetes[b] | | | |
| Yes | 196 | 7.82 (2.33) | 0.002 |
| No | 203 | 7.12 (2.02) | |

[a]Pairwise correlation

[b]ANOVA; CC: Correlation coefficient; SD: Standard deviation

significantly higher knowledge compared to the unemployed ones. There was a negative relationship between health facility level of care and DM knowledge. Participants receiving care at the community healthcare centres had a lower level of diabetes knowledge compared to those receiving care at the primary healthcare clinics.

## Discussion

Patient education and knowledge are integral parts of diabetes care. Our study found a below-average level of knowledge of diabetes among individuals living with the disease in our setting. This finding corroborates other studies within [15] and outside SA [8,10]. This indicates that

**Table 4. Ordinary least squares regression showing relationship between patients' characteristics and diabetes knowledge.**

| | Unstandardised Coefficient | Standard error | Standardised Coefficient (Beta) | p-value |
|---|---|---|---|---|
| Age | -0.01 | 0.01 | -0.07 | 0.218 |
| Gender | -0.33 | 0.28 | -0.06 | 0.229 |
| Level of education | 0.35 | 0.22 | 0.08 | 0.121 |
| Marital status | -0.37 | 0.24 | -0.08 | 0.127 |
| Employment status | 1.40 | 0.40 | 0.18 | <0.001 |
| DM family history | 0.36 | 0.22 | 0.09 | 0.100 |
| DM duration | 0.01 | 0.01 | 0.03 | 0.493 |
| Facility level | -0.77 | 0.23 | -0.17 | 0.001 |

poor knowledge is a common and persistent challenge in the management of DM. It will also impede glycaemic control and DM burden reduction. In many public health facilities in SA, diabetes education is provided by nurses, doctors and dietitians and less frequently by professional diabetes educators [26]. The situation in rural and poor settings like ours may be worse as dietitians are scarce at PHCs level and doctors are not domiciled at many of these facilities but mostly visit on outreach (primary researchers' experience). Therefore, the majority of the health education task rests on the nurses who are often overburdened with work and may also lack proper knowledge on the different components of DM management. Likewise, education materials are often produced in English and might not be culturally sensitive and this may impact knowledge transfer [26].

Knowledge of participants on the various components of DM care such as dietary practices, glucose testing, exercise, insulin use, complications identification and screening was below average. This corroborates findings from studies in Bangladesh and Pakistan that reported a low level of knowledge of DM risk factors and complications as well as the role of dietary modifications and exercise in DM management [27,28]. Of note, there was a very low knowledge on HbA1c testing which is an important component of DM care and this may be as a result of the low coverage for HbA1c testing in this setting (unpublished report). Akash et al [29] in their study conducted among DM individuals in Pakistan also reported poor knowledge of the required DM tests, the frequency, timing and the importance of the DM tests. HbA1c is a gold standard measure for glycaemic control, and this guides management plans or required treatment adjustments in order to improve outcomes. There is a need to scale up HbA1c testing and the importance of creating awareness and knowledge about it. Participants also had poor knowledge of foot care and the prevention of complications relating to DM. These are essential knowledge in order to reduce the morbidity and healthcare costs related to DM. There is a need to re-evaluate the current health education programmes at PHCs in this study setting, ensuring that the programmes include all the various components of DM management and are conducted regularly.

A positive and significant relationship was found between employment status and diabetes knowledge. This finding is in congruence with a study conducted among patients with DM in Bangladesh [10]. Employed patients might have health insurance cover which enables them to seek additional care from private health care providers where information on diabetes can be obtained. Also, employed DM patients may get more exposure through colleagues who might be sources of further information.

A higher level of education was not significantly associated with diabetes knowledge among the study participants. This contradicts findings among patients with DM in Nepal [8] and Nigeria [30]. It is expected that more educated individuals will be more inquisitive and may have the opportunity to further engage care providers in conversations relating to their health [8]. Likewise, more educated patients may be exposed to several platforms where they can obtain knowledge on diabetes and its management. This, however, did not have an impact on DM knowledge in our study and may be due to an overall low level of education of our study participants.

We found a significant and negative association between facility level and diabetes knowledge. Participants receiving care at community health centres had lesser knowledge than those at primary health care clinics. Fenwick et al [18] in their study among DM individuals in Australia showed the association between healthcare resources and DM knowledge They reported that individuals who had better access to healthcare services and personnel such as diabetes educators, counselling and support groups and DM specialists had better DM knowledge. Most of the highlighted resources by Fenwick et al [18] are absent at both the community healthcare centres and the primary healthcare clinics in this setting, except for social support

and counselling services. A plausible explanation could be that those attending primary health-care clinics in our setting had more time to engage with their care providers and attend counselling since these facilities are usually less busy than community health centres.

The mean diabetes knowledge found in this setting is below average, which is concerning and requires intervention in order to improve diabetes management and outcomes. There is already a high level of poor glycaemic control in the EC province [6], probably improving diabetes knowledge could be a positive step towards improving glycaemic control in the province. Likewise, the low level of knowledge is concerning as patients with a poor level of knowledge are least likely to comply with health instructions and regimens which impacts clinical outcomes [31].

Our study is not without limitations. We did not ascertain if the study participants had undergone any form of diabetes education which could have impacted their level of knowledge. Also, the study methodology is not appropriate for understanding the reasons for such a low level of DM knowledge. As such, future studies should adopt a mixed-methods study design or qualitative methods in exploring the reasons for low DM knowledge at PHCs level.

## Conclusion

This study provides baseline data on diabetes knowledge in Eastern Cape province where the prevalence of DM is increasing but the level of control is very low. There was a low level of knowledge on most diabetes care components, largely attributable to the employment status and the level of health facility attended by the patients with DM. Given the important role of DM knowledge on practice, complications development and glycaemic control, there is a need to design strategies for improving knowledge among DM patients in this setting, specifically targeting unemployed individuals and those attending the community healthcare centres.

## Author Contributions

**Conceptualization:** Eyitayo Omolara Owolabi, Daniel Ter Goon, Oladele Vincent Adeniyi.

**Data curation:** Eyitayo Omolara Owolabi.

**Formal analysis:** Eyitayo Omolara Owolabi, Anthony Idowu Ajayi.

**Investigation:** Eyitayo Omolara Owolabi.

**Methodology:** Eyitayo Omolara Owolabi, Anthony Idowu Ajayi, Oladele Vincent Adeniyi.

**Project administration:** Eyitayo Omolara Owolabi.

**Resources:** Eyitayo Omolara Owolabi.

**Supervision:** Eyitayo Omolara Owolabi, Daniel Ter Goon, Anthony Idowu Ajayi, Oladele Vincent Adeniyi.

**Validation:** Anthony Idowu Ajayi, Oladele Vincent Adeniyi.

**Writing – original draft:** Eyitayo Omolara Owolabi, Daniel Ter Goon, Anthony Idowu Ajayi, Oladele Vincent Adeniyi.

**Writing – review & editing:** Eyitayo Omolara Owolabi, Daniel Ter Goon, Anthony Idowu Ajayi, Oladele Vincent Adeniyi.

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
