## [Decision Letter · Decision Letter 0]

22 Feb 2022

PONE-D-21-25657Diabetes knowledge and its associated factors in rural Eastern Cape, South Africa: Findings from a cross-sectional studyPLOS ONE

Dear Dr. Ajayi,

Thank you for submitting your manuscript to PLOS ONE. After careful consideration, we feel that it has merit but does not fully meet PLOS ONE’s publication criteria as it currently stands. Therefore, we invite you to submit a revised version of the manuscript that addresses the points raised during the review process.

I have received the reports from our advisors on your manuscript which you submitted to PLOS ONE.

Based on the comments received, I feel that your manuscript could be reconsidered for publication should you be prepared to incorporate major revisions.

When preparing your revised manuscript, you are asked to carefully consider the reviewer comments below and submit a list of responses to the comments.

We look forward to receiving your revised manuscript.

Kind regards,

Muhammad Sajid Hamid Akash

Academic Editor

PLOS ONE

https://journals.plos.org/plosone/s/fileid=ba62/PLOSOne_formatting_sample_title_authors_affiliations.pdf".

Reviewers' comments:

Reviewer's Responses to Questions

**Comments to the Author**

1. Is the manuscript technically sound, and do the data support the conclusions?

Reviewer #1: Partly

2. Has the statistical analysis been performed appropriately and rigorously? 

Reviewer #1: No

3. Have the authors made all data underlying the findings in their manuscript fully available?

Reviewer #1: No

4. Is the manuscript presented in an intelligible fashion and written in standard English?

Reviewer #1: No

5. Review Comments to the Author

Reviewer #1: Knowledge of diabetes and associated factors in rural Eastern Cape, South Africa: A cross sectional study

1. In introduction, data related to pathogenesis/development of DM along with associated risk factors is missing. Try to add more data related to associated factors and prevalence of diabetes in other countries and compare them accordingly. you can take help from following references:

(J Pak Med Assoc. 2021; 71(1): 286-96. https://doi.org/10.47391/JPMA.434)

(J Biomed Sci. 2016;23(1):87. https://doi.org/10.1186/s12929-016-0303-y)

(J Biochem Mol Toxicol. 2021; https://doi.org/10.1002/jbt.22953)

(Environ Sci Pollut Res. 2020; 27(21): 26262-26275. https://doi.org/10.1007/s11356-020-09044-0)

(Clin Exp Pharmacol Physiol. 2020; 47(9):1517–1529. https://doi.org/10.1111/1440-1681.13339)

(J Cell Biochem. 2018;119(7): 5016-27. https://doi.org/10.1002/jcb.26580)

(J Cell Biochem. 2017;118(11):3577-85. https://doi.org/10.1002/jcb.26097)

2. Typing and grammatical mistakes should be corrected especially in study population and data collection.

3. In Sample and sampling technique a total of 404 participants were recruited while abstract and results are showing the data of 399 participants. Where are the remaining 5 participants?

4. In table 4, what is the contribution of marital status in diabetic knowledge? How marital status effects the knowledge of diabetes?

5. Only 6 citations are supporting your discussion, try to add more studies in your discussion and compare your results accordingly.

6. What is the aim of your study? How this study helps the health manager to strategize the knowledge about diabetes for patients as the study is not related to health care managers, it involves only diabetic patients. Clarify your conclusion

6. PLOS authors have the option to publish the peer review history of their article (what does this mean?). If published, this will include your full peer review and any attached files.

Reviewer #1: **Yes: **Sibgha Noureen

---

## [Author Response · Author response to Decision Letter 0]

21 May 2022

22 March, 2022

Dear editor,

Thank you for reviewing our manuscript entitled “Diabetes knowledge and its associated factors in rural Eastern Cape, South Africa: Findings from a cross-sectional study”, Manuscript number: PONE-D-21-25657.

Below are detailed responses to the reviewer’s comments.

Kind regards,

Anthony I. Ajayi.

Reviewer #1

Comment 1: In introduction, data related to pathogenesis/development of DM along with associated risk factors is missing. Try to add more data related to associated factors and prevalence of diabetes in other countries and compare them accordingly. you can take help from following references:

(J Pak Med Assoc. 2021; 71(1): 286-96. https://doi.org/10.47391/JPMA.434)

(J Biomed Sci. 2016;23(1):87. https://doi.org/10.1186/s12929-016-0303-y)

(J Biochem Mol Toxicol. 2021; https://doi.org/10.1002/jbt.22953)

(Environ Sci Pollut Res. 2020; 27(21): 26262-26275. https://doi.org/10.1007/s11356-020-09044-0)

(Clin Exp Pharmacol Physiol. 2020; 47(9):1517–1529. https://doi.org/10.1111/1440-1681.13339)

(J Cell Biochem. 2018;119(7): 5016-27. https://doi.org/10.1002/jcb.26580)

(J Cell Biochem. 2017;118(11):3577-85. https://doi.org/10.1002/jcb.26097)

Response: Thank you for your suggestion. We believe details about DM pathogenesis may distract the readers from the study focus but we have now added additional background information on associated risk factors as suggested.

“For instance, several studies have highlighted a positive correlation between level of education and diabetes knowledge.16-18 In addition to level of education, studies have also reported a significantly higher DM knowledge among those with a high-income level, those who had access to diabetes educators or training and among males.18 19 Others have shown the significant role of family support and being married on diabetes knowledge and practice.20”

Comment 2: Typing and grammatical mistakes should be corrected especially in study population and data collection.

Response: We have now carefully edited the manuscript for errors and mistakes.

Comment 3: In Sample and sampling technique a total of 404 participants were recruited while abstract and results are showing the data of 399 participants. Where are the remaining 5 participants?

Response: Although we recruited 404 participants, five of them were excluded at the analysis stage due to incomplete information, hence the final sample size indicated in the abstract. We have now provided this omitted explanation in the study methods:

“Prior to statistical analysis, data were checked for completeness and data from five participants were excluded from the analysis due to incomplete data.”

Comment 4: In table 4, what is the contribution of marital status in diabetic knowledge? How marital status effects the knowledge of diabetes?

Response: Previous studies have documented the significant association between presence of family support, including a marital spouse with diabetes knowledge, attitude and practice (Alaofe et al, 2021: Knowledge, attitude, practice and associated factors among patients with type 2 diabetes in Cotonou, Southern Benin; Herawati et al 2018: The Association between Knowledge, Family Support, and Blood Sugar Level in Type 2 Diabetes Mellitus Patients; Amaral et al, 2021: Factors associated with knowledge of the disease in people with type 2 diabetes mellitus). Hence, our inclusion of marital status in the bivariate analysis and the regression. Although we found a negatively significant association between marital status and DM knowledge in our bivariate analysis, this was no longer significant after adjusting for covariates.

Comment 5: Only 6 citations are supporting your discussion, try to add more studies in your discussion and compare your results accordingly.

Response: We have now added more citations and comparison. The following sentences were added:

” Knowledge of participants on the various components of DM care such as dietary practices, glucose testing, exercise, insulin use, complications identification and screening was below average. This corroborates findings from studies in Bangladesh and Pakistan that reported low level of knowledge of DM risk factors and complications as well as the role dietary modifications and exercise in DM management.26 27”

“Of note, there was a very low knowledge on HbA1c testing which is an important component of DM care and this may be as a result of the low coverage for HbA1c testing in this setting (unpublished report). Akash et al28 in their study conducted among DM individuals in Pakistan also reported poor knowledge of the required DM tests, the frequency, timing and the importance of the DM tests.”

“We found a significant and negative association between facility level and diabetes knowledge. Participants receiving care at community health centres had lesser knowledge than those at primary health care clinics. 18While this is unexpected, a plausible explanation could be that those attending primary healthcare clinics had more time to engage with their care providers as these facilities are usually less busy than community health centres. Fenwick et al 18 in their study among DM individuals in Australia showed the association between healthcare resources and DM knowledge They reported that individuals who had better access to healthcare services and personnel such as diabetes educators, counselling and support groups and DM specialists had better DM knowledge. Most of the highlighted resources by Fenwick et al18 are absent at both the community healthcare centres and the primary healthcare clinics in this setting, except for social support and counselling services. A plausible explanation could be that those attending primary healthcare clinics in our setting had more time to engage with their care providers and attend counselling since these facilities are usually less busy than community health centres.”

Comment 6: What is the aim of your study? How this study helps the health manager to strategize the knowledge about diabetes for patients as the study is not related to health care managers, it involves only diabetic patients. Clarify your conclusion

Response: We have revised and adjusted this section of the manuscript. It now reads:

“Given the important role of DM knowledge on practice, complications development and glycaemic control, there is a need to design strategies for improving knowledge among DM patients in this setting, specifically targeting unemployed individuals and those attending the community healthcare centres.”

---

## [Editor Report · Decision Letter 1]

30 May 2022

Knowledge of diabetes and associated factors in rural Eastern Cape, South Africa: findings from a multicentre cross-sectional study

PONE-D-21-25657R1

Dear Dr. Ajayi,

We’re pleased to inform you that your manuscript has been judged scientifically suitable for publication and will be formally accepted for publication once it meets all outstanding technical requirements.

Kind regards,

Muhammad Sajid Hamid Akash

Academic Editor

PLOS ONE
---

## [Editor Report · Acceptance letter]

7 Jul 2022

PONE-D-21-25657R1 

Knowledge of diabetes and associated factors in rural Eastern Cape, South Africa: A cross sectional study 

Dear Dr. Ajayi:

I'm pleased to inform you that your manuscript has been deemed suitable for publication in PLOS ONE. Congratulations! Your manuscript is now with our production department. 

Kind regards, 

on behalf of

Dr. Muhammad Sajid Hamid Akash 

Academic Editor

PLOS ONE